# Noise-Aware Statistical Inference with Differentially Private Synthetic Data

**Ossi Räisä**[*]   **Joonas Jälkö**[†]   **Samuel Kaski**[†‡]   **Antti Honkela**[*]

[*]Helsinki Institute for Information Technology HIIT, Department of Computer Science,
University of Helsinki

[†]Helsinki Institute for Information Technology HIIT, Department of Computer Science,
Aalto University

[‡]Department of Computer Science, University of Manchester

`{ossi.raisa, joonas.jalko, antti.honkela}@helsinki.fi samuel.kaski@aalto.fi`

## Abstract

Existing work has shown that analysing differentially private (DP) synthetic data as if it were real does not produce valid uncertainty estimates. We tackle this problem by combining synthetic data analysis techniques from the field of multiple imputation (MI), and synthetic data generation using a novel noise-aware (NA) synthetic data generation algorithm NAPSU-MQ into a pipeline NA+MI that allows computing accurate uncertainty estimates for population-level quantities from DP synthetic data. Our experiments demonstrate that the pipeline is able to produce accurate confidence intervals from DP synthetic data.

## 1 Introduction

Availability of data for research is constrained by the dilemma between privacy preservation and potential gains obtained from sharing. One solution to the dilemma is to have the *data holder* release a synthetic dataset based on a real dataset. *Data analysts* can then use the synthetic data for their *downstream analysis*. Privacy-protection of the synthetic data can be guaranteed by employing differential privacy (DP) [10], which offers provable protection.

While many works have studied synthetic data generation under DP [2, 4, 6, 7, 8, 12, 13, 16, 19, 20, 21, 23, 24, 25, 26, 28, 35, 37, 38, 39], the analysis of synthetic data has received far less attention. This is especially apparent for uncertainty estimates, like $p$-values and confidence intervals. It is clear that synthetic data generation adds additional uncertainty [29], especially under DP, which requires adding extra noise. However, only two of the mentioned works [25, 28] consider uncertainty estimation of query values on the real data, and none of them consider uncertainty estimation of population parameters.

We illustrate the issue with treating synthetic data as if it were real using a simple toy data experiment. We generate 3-dimensional binary data, where one variable is generated from logistic regression on the other two, serving as the original dataset. Then, we generate synthetic data from the original data and compute confidence intervals for the coefficients of the logistic regression from the synthetic data. We describe the setting in more detail in Supplementa Section F As shown by Figure 1, the existing algorithms PGM [26], PEP [20], and RAP [2] do not provide valid confidence intervals, as they treat the synthetic data as real data. Only our method NA+MI and PrivLCM [28] are able provide valid confidence intervals. However, PrivLCM is too conservative and produces much wider intervals than NA+MI, as shown by Figure 3a.

NeurIPS 2022 Workshop on Synthetic Data for Empowering ML Research.

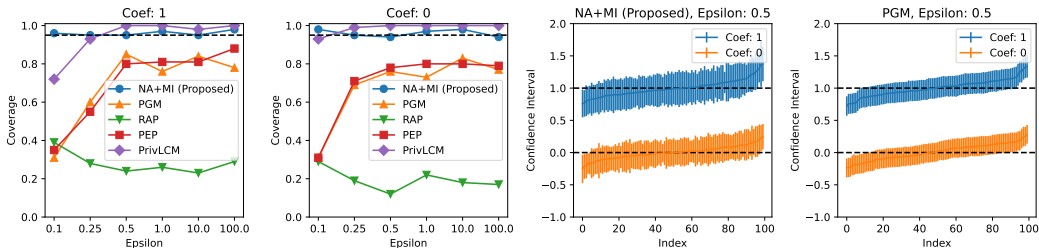

Figure 1: Toy data experiment results of logistic regression on 3 binary variables, showing that all algorithms apart from ours and PrivLCM are overconfident, even with almost no privacy ($\epsilon = 100$). The first two panels from the left show the fraction of the 95% confidence intervals that contain the true parameter value in 100 repeats, with the target confidence level of 95% highlighted by a black line. The third panel shows the confidence intervals from synthetic data generated by our mechanism for $\epsilon = 0.5$, and the fourth panel shows the confidence intervals from PGM. The third and fourth panels show that the overconfidence stems from intervals that are too narrow, a result of not accounting for all uncertainty.

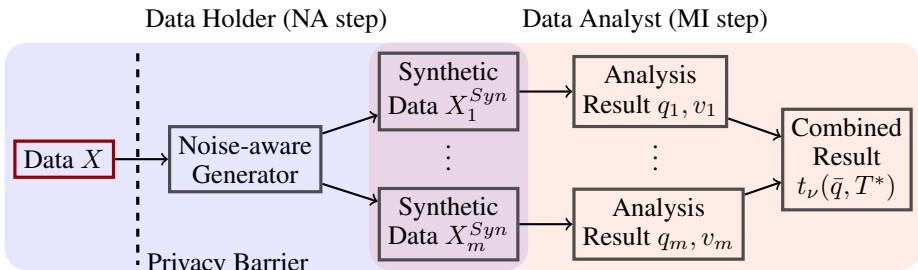

Figure 2: NA+MI pipeline for noise-aware DP synthetic data generation and statistical inference. The nodes shaded in blue are computed by the data holder, and the nodes shaded in orange are computed by the data analyst. All nodes except Data (with red border) can be released to the public. The synthetic datasets can be generated by either party because the Generator is also released by the data holder.

## 2 The NA+MI Pipeline

To compute accurate uncertainty estimates from DP synthetic data, we build on the framework of Rubin [32]. He proposed generating multiple datasets from a posterior predictive distribution, and using simple combining rules called Rubin's rules to account for the extra uncertainty that synthetic data generation introduces. This workflow is modeled after *multiple imputation* (MI), where it is used to deal with missing data. We summarise Rubin's rules in more detail in Supplemental Section B.

Rubin's rules are able to account for the extra noise added by DP when the posterior predictive distribution that generates the synthetic data accurately reflects the noise addition process. We call Bayesian inference algorithms that account for DP noise *noise-aware* (NA).

Figure 2 summarises our proposed pipeline, which we call NA+MI. First, the data holder runs inference on a noise-aware Bayesian model using the private data, which we call the NA step. After the inference, the data holder generates multiple synthetic datasets. The data holder can also release the posterior distribution in addition to the synthetic datasets, so that the analyst can also generate synthetic datasets if needed.

For each synthetic dataset, the analyst runs their analysis, producing a point estimates $q_i$ and a variance estimate $v_i$. These are combined with Rubin's rules, which produce a $t$-distribution that the analyst can use to compute confidence intervals or hypothesis tests.

# 3 Noise-aware Synthetic Data Generation

To implement the NA step, we develop an algorithm called *Noise-Aware Private Synthetic data Using Marginal Queries* (NAPSU-MQ), that generates synthetic data from discrete tabular datasets using the noisy values of preselected marginal queries.

**Data and Marginal Queries**   NAPSU-MQ uses tabular datasets of $d$ discrete variables, where the domains of the discrete variables, as well as the number datapoints $n$ are known. We denote the set of possible datapoints by $\mathcal{X}$, and the set of possible datasets by $\mathcal{X}^n$.

**Definition 3.1.** *A marginal query of variables $I$ and value $x$ is a function $a\colon \mathcal{X} \to \{0,1\}$ that takes a datapoint as input and returns 1 if the variables in $I$ in the datapoint have the value $x$, and 0 otherwise. For a dataset $X \in \mathcal{X}^n$, we define $a(X) = \sum_{i=1}^n a(X_i)$, where $X_i$ is the i:th datapoint in $X$.*

When $I$ has $k$ variables, $a$ is called a $k$-way marginal query.

When evaluating multiple marginal queries $a_1, \ldots, a_{n_q}$, we concatenate their values to a vector-valued function $a\colon \mathcal{X} \to \{0,1\}^{n_q}$. We call the concatenation of marginal queries for all possible values of variables $I$ the *full set of marginals on $I$* [1].

**Implemetation**   In order to implement the NA step, the data holder needs to generate synthetic data from the posterior of a noise-aware Bayesian model. Bernstein and Sheldon [5] develop noise-aware Bayesian inference under DP for simple exponential family models. We implement the NA step by generalising their algorithm to arbitrary marginal queries, using the maximum entropy distribution of the marginal queries as our exponential family model.

The distribution with maximal entropy with the correct expected marginal query values is

$$P(x) = \exp(\theta^T a(x) - \theta_0(\theta)) \tag{1}$$

for some parameters $\theta \in \mathbb{R}^k$ [36], where $k$ is the number of queries. $\theta_0(\theta)$ is the normalising constant of the distribution. We denote this distribution by $\mathrm{MED}_\theta$, and use $\mathrm{MED}_\theta^n$ to denote the distribution of $n$ i.i.d. samples from $\mathrm{MED}_\theta$.

The Bayesian model we consider is derived from the generative process of the noisy query values, which are observed. Assuming that the data generating process is $\mathrm{MED}_\theta$, and knowing that the Gaussian mechanism adds noise with variance $\sigma_{DP}^2$, the probabilistic model generating $\tilde{s}$ is

$$\theta \sim \mathrm{Prior}, \quad X \sim \mathrm{MED}_\theta^n, \quad s = a(X), \quad \tilde{s} \sim \mathcal{N}(s, \sigma_{DP}^2 I). \tag{2}$$

In principle, we could now sample from the posterior $p(\theta \mid \tilde{s})$, with $s$ marginalised out. In practice, the marginalisation is not feasible, as $s$ is a discrete variable with a very large domain.

However, $s$ is a sum of the query values for individual datapoints, so asymptotically $s$ has a normal distribution according to the central limit theorem. We can substitute the normal approximation for $s$ into the model, which allows us to easily marginalise $s$ out, resulting in the approximation

$$\theta \sim \mathrm{Prior}, \quad \tilde{s} \sim \mathcal{N}(n\mu(\theta), n\Sigma(\theta) + \sigma_{DP}^2 I), \tag{3}$$

where $\mu(\theta)$ and $\Sigma(\theta)$ denote the mean and covariance of $a(x)$ for a single sample $x \sim \mathrm{MED}_\theta$.

$\mu(\theta)$ and $\Sigma(\theta)$ can be computed efficiently using techniques from probabilistic graphical models and autodifferentiation when the graph corresponding to the selected queries is sparse. We can then compute the approximate unnormalised density, which allows us to use off-the-shelf sampling algorithms, like the Laplace approximation or the No-U-Turn sampler (NUTS) [14].

For the prior, we choose another Gaussian distribution with mean 0 and standard deviation 10, which is a simple and weak prior, but other priors could be used.

We can make $\mathrm{MED}_\theta$ identifiable by pruning the queries using the *canonical parameterisation* [18] of $\mathrm{MED}_\theta$, which we describe in more detail in Supplemental Section C. Algorithm 1 summarises NAPSU-MQ.

---

[1] Some existing works [26] use the term marginal query for the full set of marginal queries. We chose this terminology because we deal with individual marginal queries in Supplemental Section C.

**Algorithm 1:** NAPSU-MQ

---

**Input:** Real data $X$, marginal queries $a$, number of synthetic datasets $m$, size of synthetic datasets $n_{Syn}$, privacy bounds $\epsilon, \delta$.
**Output:** Posterior distribution $p(\theta|\tilde{s})$, synthetic datasets $X_1^{Syn}, \dots, X_m^{Syn}$.
$a^* \leftarrow$ Canonical queries for $a$ (Section C);
$s \leftarrow a^*(X)$;
$\Delta_2 \leftarrow$ Sensitivity of $s$ (Theorem A.2);
$\sigma_{DP}^2 \leftarrow$ Required noise variance for $(\epsilon, \delta)$-DP with sensitivity $\Delta_2$ (Theorem A.3);
Sample $\tilde{s} \sim \mathcal{N}(s, \sigma_{DP}^2)$;
Run Bayesian inference algorithm to find $p(\theta|\tilde{s})$ (Section 3);
Sample $\theta_i \sim p(\theta|\tilde{s})$ and $X_i^{Syn} \sim \text{MED}_{\theta_i}^{n_{Syn}}$ for $1 \le i \le m$;
**return** $p(\theta|\tilde{s})$, $X_1^{Syn}, \dots, X_m^{Syn}$

---

## 4 Privacy of NAPSU-MQ

To quantify privacy, we use differential privacy (DP) [9, 10], which aims to rigorously quantify the privacy loss resulting from releasing the results of an algorithm. DP algorithms are also called *mechanisms*. This section presents an overview of the privacy of NAPSU-MQ. The more technical details are in Supplemental Section A.

**Definition 4.1.** *A mechanism $\mathcal{M}$ is $(\epsilon, \delta)$-differentially private if for all neighbouring datasets $X, X'$ and all measurable output sets $S$*

$$P(\mathcal{M}(X) \in S) \le e^\epsilon P(\mathcal{M}(X') \in S) + \delta. \tag{4}$$

The neighbourhood relation in the definition is domain specific. We use the *substitute* neighbourhood relation for tabular datasets, where datasets are neighbouring if they differ in at most one datapoint.

The mechanism we use to release marginal query values under DP is the *Gaussian mechanism* [9].

**Definition 4.2.** *The Gaussian mechanism with noise variance $\sigma^2$ adds Gaussian noise to the value of a function $f: \mathcal{X}^n \to \mathbb{R}^k$ for input data $X$: $\mathcal{M}(X) = f(X) + \mathcal{N}(0, \sigma^2 I)$.*

An important property of DP is post-processing immunity: post-processing the result of a DP-algorithm does not weaken the privacy bounds.

**Theorem 4.3** (Dwork and Roth [11])**.** *Let $\mathcal{M}$ be an $(\epsilon, \delta)$-DP mechanism, and let $f$ be any algorithm. Then the composition $f \circ \mathcal{M}$ is $(\epsilon, \delta)$-DP.*

Because of post-processing immunity, we can release marginal query values with the Gaussian mechanism to obtain privacy bounds, and make arbitrary use of the noisy query values without weakening the privacy bounds. This makes the privacy proof of NAPSU-MQ very simple:

**Theorem 4.4.** *NAPSU-MQ (Algorithm 1) is $(\epsilon, \delta)$-DP with regards to the real data $X$.*

*Proof.* The returned values of Algorithm 1 only depend on the real data $X$ through $\tilde{s}$. Releasing $\tilde{s}$ is $(\epsilon, \delta)$-DP due to the selection of $\sigma_{DP}^2$ with Theorem A.3. Computing the returned values from $\tilde{s}$ is post-processing, so by Theorem 4.3, NAPSU-MQ is $(\epsilon, \delta)$-DP. $\qquad\square$

## 5 Experiments

In this section, we summarise our experiments and their results. We give detailed descriptions of hyperparameters in Supplemental Section E, of the toy data experiment in Supplemental Section F, and of the Adult experiment in Supplemental Section G We describe an additional experiment with the UCI US Census (1990) dataset in Supplemental Section H.

**Ablation Study** In addition to the comparison presented in Figure 1, we conducted an ablation study on the same toy data setting to show that both multiple imputation and noise-awareness are necessary for accurate confidence intervals. The results are presented in Figure 3b. Without both multiple imputation and noise-awareness, NA+MI is overconfident like PGM, except for $\epsilon \ge 1$, where noise-awareness is not required.

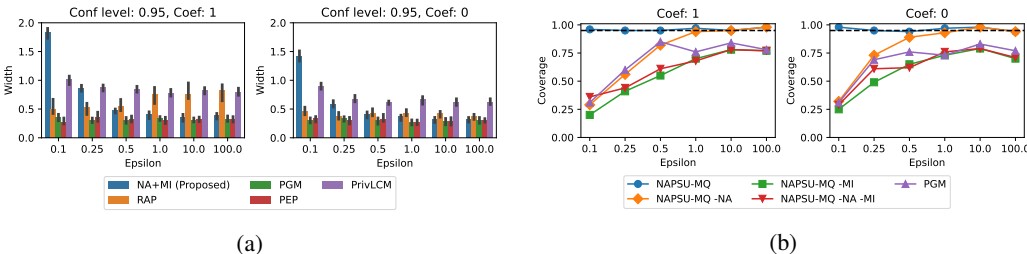

Figure 3: (a) Toy data confidence intervals widths. NA+MI produces slightly wider intervals than PGM, or PEP for $\epsilon > 0.1$, which is necessary to account for DP noise. PrivLCM produces much wider intervals. (b) Ablation study on the toy data. "-NA" refers to removing noise-awareness, and "-MI" refers to removing multiple imputation. Unless both are included, NAPSU-MQ is overconfident like PGM except for $\epsilon \geq 1$ where noise-awareness is not necessary.

## 5.1    Adult Dataset

Our main experiment evaluates the performance of NAPSU-MQ on the UCI Adult dataset [17]. We include 10 of the original 15 columns to remove redundant columns and keep runtimes manageable, and discretise the continuous columns. After dropping rows with missing values, there are $n = 46\,043$ rows. The discretised domain has $1\,792\,000$ distinct values.

We compare NAPSU-MQ against PGM [26], RAP [2] and PEP [20]. We used the published implementations of their authors for all of them, with small modifications to ensure compatibility with new library versions and our experiments. The published implementation of PrivLCM only supports binary data, and does not scale to datasets of this size, so it was not included in this experiment. We also include a naive noise-aware baseline that runs $m$ completely independent repeats of PGM, splitting the privacy budget appropriately, and uses Rubin's rules with the $m$ generated synthetic datasets.

Reiter [30] discusses the choice of $n_{Syn}$ and $m$ for non-DP synthetic data generation in detail. Based on his results, choosing $n_{Syn} = n$, is very safe, and we use it for all algorithm except RAP, where $n_{Syn}$ is a significant hyperparameter. For NAPSU-MQ and PGM-repeat, we choose the number of generated synthetic datasets with a preliminary experiment, presented in Figures S4 and S3. For NAPSU-MQ, Reiter's theory [30] suggests that larger $m$ is better, with diminishing returns, which is supported by our experiment.

As our downstream task, we use logistic regression with income as the dependent variable, and a subset of the columns as independent variables, which allows us to include all the relevant marginals for synthetic data generation. Note that the synthetic dataset was still generated with all 10 columns.

NAPSU-MQ and PGM-repeat sometimes generate synthetic datasets with no people of some race with high income. This causes the downstream logistic regression to produce an extremely wide confidence interval for the coefficient of that race. As Rubin's rules average over estimates and estimated variances, even a single one of these bad estimates makes the combined confidence interval extremely wide. To fix this issue, we can either remove the estimates with estimated standard deviations greater than $10^{1.5}$ before applying Rubin's rules, or add a small regularisation term to the logistic regression, and use bootstrapping for variance estimates. Both methods produce almost identical results, but bootstrapping significantly increases the computational cost of the downstream analysis.

As the input queries, we pick 2-way marginals that are relevant for the downstream task, and select the rest of the queries with the MST algorithm [24]. This selection was kept constant throughout the experiment. For the privacy budget, we use $\delta = n^{-2}$ for all runs, and vary $\epsilon$.

The Laplace approximation for NAPSU-MQ does not work well for this setting because many of the queries have small values, so we use NUTS [14] for posterior inference. To speed up NUTS, we normalise the posterior before running the inference using the mean and covariance of the Laplace approximation

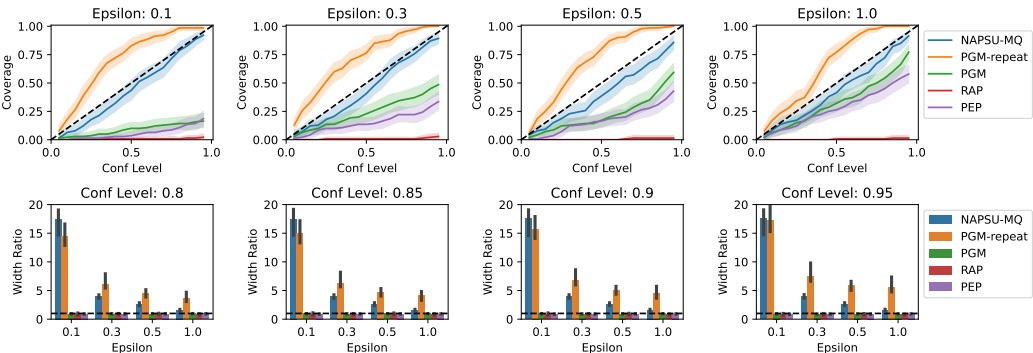

Figure 4: Top row: the fraction of downstream coefficients where the synthetic confidence interval contains the real data coefficient, averaged over 20 repeated runs. NAPSU-MQ is the only algorithm that is consistently around the diagonal, showing good calibration. The error bands are bootstrap 95% confidence intervals of the average. Bottom row: confidence intervals widths divided by real data confidence interval widths. Each bar is a median of medians over the different coefficients and repeats, and the black lines are 95% bootstrapped confidence intervals. The dashed line is at $y = 1$, showing where synthetic confidence intervals have the same width as original confidence intervals.

Results from 20 repeats of the experiment are shown in Figure 4. PGM, RAP and PEP produce overconfident confidence intervals that do not meet the given confidence levels. With the repeats, PGM becomes overly conservative, and produces confidence intervals that are too wide. NAPSU-MQ is the only algorithm that produces properly calibrated intervals, although repeated PGM is able to produce narrower intervals than NAPSU-MQ with $\epsilon = 0.1$. With the more realistic $\epsilon = 1$, the confidence intervals from NAPSU-MQ are not much wider than non-DP intervals, while PGM-repeat produces much wider intervals. Figure S8 shows that NAPSU-MQ reproduces 1- and 2-way marginals nearly as accurately as PGM.

The results in Figure 4 were obtained using the small regularisation term. Figure S5 shows the results with the trick of dropping large variances, which are very close to Figure 4.

Noise-awareness, especially with the increased accuracy from NUTS, comes with a steep computational cost, as PGM ran in 15s, while the Laplace approximation took several minutes, and NUTS took up to ten hours. All of the algorithms were run on 4 CPU cores of a cluster. The complete set of runtimes for all algorithms and values of $\epsilon$ are shown in Table S1 of the Supplement.

## 6    Discussion

While our general pipeline NA+MI is applicable to all kinds of datasets in principle, the data generation algorithm NAPSU-MQ is currently only applicable to discrete tabular data and only supports sparse marginal queries perturbed with the Gaussian mechanism as input. We aim to generalise NAPSU-MQ to more general query classes, such as linear queries, in the future, but supporting other types of noise is likely much harder. Only handling discrete data is not a major limitation, as discretisation algorithms [40] have been shown to perform very well on tabular synthetic data generation tasks [34].

The Gaussian mechanism adds noise uniformly to all queries, making small queries relatively more noisy. This may reduce the accuracy of query-based algorithms like NAPSU-MQ and the others we examined for groups with rare combinations of data values, such as minorities.

Although we left query selection outside the scope of this paper, selecting the right queries to support downstream analysis is important, as NA+MI cannot guarantee confidence interval coverage if the selected queries do not contain enough information for the downstream task. We plan to study whether existing methods giving confidence bounds on query accuracy [25, 28] can be adapted to give confidence intervals for arbitrary downstream analyses.

The runtime of NAPSU-MQ, especially when using NUTS, is another major limitation. As NAPSU-MQ is compatible with any non-DP posterior sampling method, recent [15] and future advances in MCMC and other sampling techniques are likely able to cut down on the runtime.

**Conclusion**   The analysis of DP synthetic data has not received much attention in existing research. Our work patches a major hole in the current generation and analysis methods by developing the NA+MI pipeline that allows computing accurate confidence intervals and $p$-values from DP synthetic data. We develop the NAPSU-MQ algorithm in order to implement NA+MI on nontrivial discrete datasets. While NAPSU-MQ has several limitations, NA+MI only depends on noise-aware posterior inference, not NAPSU-MQ specifically, and can thus be extended to other settings in the future. With the noise-aware inference algorithm, NA+MI allows conducting valid statistical analyses that include uncertainty estimates with DP synthetic data, potentially unlocking existing privacy-sensitive datasets for widespread analysis.

## Acknowledgements

This work was supported by the Academy of Finland (Flagship programme: Finnish Center for Artificial Intelligence, FCAI; and grants 325572, 325573), the Strategic Research Council at the Academy of Finland (Grant 336032) as well as UKRI Turing AI World-Leading Researcher Fellowship, EP/W002973/1. The authors wish to thank the Finnish Computing Competence Infrastructure (FCCI) for supporting this project with computational and data storage resources.

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

# Suppelementary Material for Noise-Aware Statistical Inference with Differentially Private Synthetic Data

## A    Omitted Privacy Results

The privacy bounds of the Gaussian mechanism depend on the *sensitivity* of the function $f$, which is an upper bound on the change in the value of $f$ for neighbouring datasets.

**Definition A.1.** *The $L_2$-sensitivity of a function $f$ is $\Delta_2 f = \sup_{X \sim X'} ||f(X) - f(X')||_2$. $X \sim X'$ denotes that $X$ and $X'$ are neighbouring.*

**Theorem A.2.** *Let $a$ be the concatenation of $n_s$ full sets of marginal queries. Then $\Delta_2 a \leq \sqrt{2n_s}$.*

*Proof.* Let $a_1, \ldots, a_{n_s}$ be the full sets of marginal queries that form $a$. Because all of the queries of $a_i$ have the same set of variables, the vector $a_i(x)$ has a single component of value 1, and the other components are 0 for any $x \in \mathcal{X}$. Then, for any neighbouring $X, X' \in \mathcal{X}^n$, $||a_i(X) - a_i(X')||_2^2 \leq 2$. Then

$$\Delta_2 a = \sup_{X \sim X'} ||a(X) - a(X')|| \tag{5}$$

$$= \sup_{X \sim X'} \sqrt{\sum_{i=1}^{n_s} ||a_i(X) - a_i(X')||_2^2} \tag{6}$$

$$\leq \sup_{X \sim X'} \sqrt{\sum_{i=1}^{n_s} 2} \tag{7}$$

$$= \sqrt{2n_s} \tag{8}$$

□

**Theorem A.3** (Balle and Wang [3]). *The Gaussian mechanism for function $f$ with $L_2$-sensitivity $\Delta_2$ and noise variance $\sigma^2$ is $(\epsilon, \delta)$-DP with*

$$\delta \geq \Phi\left(\frac{\Delta_2}{2\sigma} - \frac{\epsilon\sigma}{\Delta_2}\right) - e^\epsilon \Phi\left(-\frac{\Delta_2}{2\sigma} - \frac{\epsilon\sigma}{\Delta_2}\right) \tag{9}$$

*where $\Phi$ is the cumulative distribution function of the standard Gaussian distribution.*

## B    Multiple Imputation

In order to compute uncertainty estimates for downstream analyses from the noise-aware posterior with NA+MI, we use Rubin's rules for synthetic data [29, 30].

After the synthetic datasets $X_i^{Syn}$ for $1 \leq i \leq m$ are released by the data holder, the data analyst runs their downstream analysis on each $X_i^{Syn}$. For each synthetic dataset, the analysis produces a point estimate $q_i$ and a variance estimate $v_i$ for $q_i$.

The estimates $q_1, \ldots, q_m$ and $v_1, \ldots, v_m$ are combined as follows [29]:

$$\bar{q} = \frac{1}{m} \sum_{i=1}^{m} q_i, \quad \bar{v} = \frac{1}{m} \sum_{i=1}^{m} v_i, \quad b = \frac{1}{m-1} \sum_{i=1}^{m} (q_i - \bar{q})^2. \tag{10}$$

We use $\bar{q}$ as the combined point estimate, and set

$$T = \left(1 + \frac{1}{m}\right) b - \bar{v}, \qquad T^* = \begin{cases} T & \text{if } T \geq 0 \\ \frac{n_{Syn}}{n} \bar{v} & \text{otherwise.} \end{cases} \tag{11}$$

$T$ is an estimate of the combined variance. $T$ can be negative, which is corrected using $T^*$ instead [30].

We compute confidence intervals and hypothesis tests using the $t$-distribution with mean $\bar{q}$, variance $T^*$, and degrees of freedom

$$\nu = (m-1)(1 - r^{-1})^2, \tag{12}$$

where $r = (1 + \frac{1}{m})\frac{b}{v}$ [30].

These combining rules apply when $q$ is a univariate estimate. Reiter [31] derives appropriate combining rules for multivariate estimates, which can be applied with NA+MI.

Rubin's rules make many assumptions on the different distributions that are involved [29, 33], such as the normality of the distribution of $q_i$ when sampling data from the population. These assumptions may not hold for some types of estimates, such as probabilities [22] or quantile estimates [41]. Further work [33] tries to reduce these assumptions, especially in the context of missing data. Their results for synthetic data generation can be applied with our method.

Si and Reiter [33] propose to remove some of these assumptions by approximating the integral that Rubin's rules are derived from by sampling instead of using the analytical approximations in (10) and (11). They find that their sampling-based approximation can be effective, especially with a small number of datasets, but is computationally more expensive.

## B.1 Unbiasedness of Rubin's Rules

Rubin's rules make several assumptions on the downstream analysis method, and several normal approximations when deriving the rules. Raghunathan, Reiter, and Rubin [29] derive conditions under which Rubin's rules give an unbiased estimate.

Rubin's rules aim to estimate a quantity $Q$ of the entire population $P$, of which $X$ is a sample. Conceptually, the sampling of the synthetic datasets is done in two stages: first, synthetic populations $P_i^{Syn}$ for $1 \leq i \leq m$ are sampled. Second, a synthetic dataset $X_i^{Syn}$ is sampled i.i.d from $P_i^{Syn}$. This is equivalent to the sampling process for $X_i^{Syn}$ described in Section 2, and makes stating the assumptions of Rubin's rules easier.

While the sampling process of synthetic data has to be i.i.d., this is not required for the original data. This means that there are two versions of the downstream analysis: the one for i.i.d. data, and the one for the complex sampling method of the real data. The latter method is not used at any point, so it is assumed to exist for the theory, but does not have to be practically implementable.

We take advantage of this handling of complex sampling of real data by including the computation of $s$ and adding noise to get $\tilde{s}$ in the sampling scheme, so we are considering $\tilde{s}$ to be the original data from the point of view of the theory. The fact that $\tilde{s}$ is a noisy summary statistic instead of a dataset is not an issue, as the theory only requires having a theoretical method to estimate $Q$ from $\tilde{s}$. If the chosen marginal queries contain the relevant information for the downstream task, so $s$ is a (approximate) sufficient statistic, this theoretical method will exist.

Let $Q_i$ denote the quantity of interest $Q$ computed from the synthetic population $P_i^{Syn}$ instead of $P$. Let $V_i$ denote the sampling variance of $q_i$ from the synthetic population $P_i^{Syn}$. Note that $q_i$ and it's variance estimate $v_i$ are obtained using the downstream analysis method for i.i.d. data. Let $\hat{Q}_D$ and $\hat{U}_D$ be the point and variance estimates of $Q$ derived from $\tilde{s}$ when sampling the population $P$, which are obtained using the theoretical inference method for complex samples.

Now the assumptions of Raghunathan, Reiter, and Rubin [29] are

**Assumption B.1.** *For all $1 \leq i \leq m$, $q_i$ is unbiased for $Q_i$ and asymptotically normal with respect to sampling from the synthetic population $P_i^{Syn}$, with sampling variance $V_i$.*

**Assumption B.2.** *For all $1 \leq i \leq m$, $v_i$ is unbiased for $V_i$, and the sampling variability in $v_i$ is negligible. That is $v_i \mid P_i^{Syn} \approx V_i$. Additionally, the variation in $V_i$ across the synthetic populations is negligible.*

Assumptions B.1-B.2 ensure that the downstream analysis method used to estimate $Q$ is accurate, for both point and variance estimates, when applied to i.i.d. real data, regardless of the population.

**Assumption B.3.** $\hat{Q}_D \mid P \sim \mathcal{N}(Q, \hat{U}_D)$

Assumption B.3 ensures that the analysis for complex sampling is accurate for point and variance estimates when applied to the real population.

**Assumption B.4.** $Q_i \mid \tilde{s} \sim \mathcal{N}(\hat{Q}_D, \hat{U}_D)$

Assumption B.4 requires that the generation of synthetic datasets does not bias the downstream analysis.

For query-based methods like NAPSU-MQ, Assumptions B.3 and B.4 may not hold when the queries do not contain the relevant information for the downstream task.

With Assumptions B.1-B.4, Raghunathan, Reiter, and Rubin [29] show that $\bar{q}$ is an unbiased estimate of $Q$, and $T$ is an asymptotically unbiased variance estimate.

**Theorem B.5** (Raghunathan, Reiter, and Rubin [29]). *Assumptions B.1-B.4 imply that*

1. $E(\bar{q} \mid P) = Q$,

2. $E(T \mid P) = \mathrm{Var}(\bar{q} \mid P)$,

3. *Asymptotically* $\frac{\bar{q}-Q}{\sqrt{T}} \sim \mathcal{N}(0, 1)$,

4. *For moderate* $m$, $\frac{\bar{q}-Q}{\sqrt{T}} \sim t_\nu(0, 1)$ *[30].*

## C  Finding an Identifiable Parametrisation

In this section, we describe the process we use to ensure the parametrisation of the posterior in NAPSU-MQ is identifiable. We ensure identifiability by dropping some of the selected queries, chosen using the the canonical parametrisation of $\mathrm{MED}_\theta$ to ensure no information is lost. First, we give some background on Markov networks, which is necessary to understand the canonical parametrisation.

**Markov Networks**   A Markov network is a representation of a probability distribution that is factored according to an undirected graph. Specifically, a Markov network distribution $P$ is a product of *factors*. A factor is a function from a subset of the variables to non-negative real numbers. The subset of variables is called the *scope* of the factor. The joint distribution is given by

$$P(x) = \frac{1}{Z} \prod_{I \subset S} \phi_I(x_I) \tag{13}$$

where $S$ is the set of scopes for the factors. The undirected graph is formed by representing each variable as a node, and adding edges such that the scope of each factor is a clique in the graph.

**Canonical Parametrisation**   The canonical parametrisation is given in terms of *canonical factors* [1]. The canonical factors depend on an arbitrary assignment of variables $x^*$. We simply choose $x^* = (0, \ldots, 0)$. In the following, $x_U$ denotes the selection of components in the set $U$ from the vector $x$, and $x_{-U}$ denotes the selection of all components except those in $U$.

**Definition C.1.** *A canonical factor $\phi_D^*$ with scope $D$ is defined as*

$$\phi_D^*(x) = \exp\left(\sum_{U \subseteq D} (-1)^{|D-U|} \ln P(x_U, x_{-U}^*)\right)$$

*The sum is over all subsets of $D$, including $D$ itself and the empty set. $|D - U|$ is the size of the set difference of $D$ and $U$.*

**Theorem C.2** (Abbeel, Koller, and Ng [1](Theorem 3)). *Let $P$ be a Markov network with factor scopes $S$. Let $S^* = \cup_{D \in S} \mathcal{P}(D) - \emptyset$. Then*

$$P(x) = P(x^*) \prod_{D^* \in S^*} \phi_{D^*}^*(x_{D^*})$$

There are more canonical factors than original factors, so it might seem that there are more parameters in the canonical parametrisation than in the original parametrisation. However, many values in the canonical factors turn out to be ones. We can select the queries corresponding to non-one canonical factor values to obtain a set of queries with the same information as the original queries, but without linear dependencies [18]. We call this set of queries the *canonical queries*.

Many of the canonical factor scopes are subsets of the original factor scopes, so using the canonical queries as is would introduce new marginal query sets and potentially increase the sensitivity of the queries. As all of the new queries are sums of existing queries, we can replace each new query with the old queries that sum to the new query, and use the same $\theta$ value for all of the added queries to preserve identifiability. If one of the added queries was already included, it does not need to be added again, because two instances of a single query can be collapsed into a single instance with it's own parameter value. Because of this, we did not need to fix the $\theta$ values of any queries to the same value in the settings we studied.

## D  NAPSU-MQ vs. PGM

The PGM algorithm [26] generates synthetic data based on the same marginal queries $a$ and noise addition as NAPSU-MQ. PGM also models the original data using the $\mathrm{MED}_\theta$ distribution. Unlike NAPSU-MQ, PGM finds the parameters $\theta$ by minimising the $l_2$-distance $||\tilde{s} - n\mu(\theta)||_2$ between the observed noisy query values $\tilde{s}$ and the expected query values $n\mu(\theta) = nE_{x \sim \mathrm{MED}_\theta}(a(x))$ In the following, we'll replace the query values $s$ and $\tilde{s}$ that are summed over datapoints with $u = \frac{s}{n}$ and $\tilde{u} = \frac{\tilde{s}}{n}$ that represent mean query values over datapoints. Then the PGM objective is equivalent to $||\tilde{u} - \mu(\theta)||_2$.

We can view the PGM minimisation problem as a maximum likelihood estimation in the NAPSU-MQ probabilistic model

$$X \sim \mathrm{MED}_\theta^n, \quad s = a(X), \quad \tilde{s} \sim \mathcal{N}(s, \sigma_{DP}^2 I), \tag{14}$$

where we replace normal approximation that NAPSU-MQ uses with a law of large numbers approximation. Specifically, first replace $s$ with $u$ in (14):

$$X \sim \mathrm{MED}_\theta^n, \quad u = \frac{a(X)}{n}, \quad \tilde{u} \sim \mathcal{N}(u, \sigma_{DP}^2 I/n^2). \tag{15}$$

Because $u$ is a mean of sufficient statistics for individual datapoints, by the law of large numbers, asymptotically $u \sim \delta_{\mu(\theta)}$. With this approximation, the probabilistic model is

$$u \sim \delta_{\mu(\theta)}, \quad \tilde{u} \sim \mathcal{N}(u, \sigma_{DP}^2 I/n^2). \tag{16}$$

$u$ can be marginalised from the likelihood of this model:

$$p(\tilde{u}|\theta) = \int p(\tilde{u}, u|\theta)\mathrm{d}u \tag{17}$$

$$= \int p(\tilde{u}|u)p(u|\theta)\mathrm{d}u \tag{18}$$

$$= \int \mathcal{N}(\tilde{u}|u, \sigma_{DP}^2 I/n^2)\delta_{\mu(\theta)}(u)\mathrm{d}u \tag{19}$$

$$= \mathcal{N}(\tilde{u}|\mu(\theta), \sigma_{DP}^2 I/n^2) \tag{20}$$

The marginalised log-likelihood is then

$$\ln p(\tilde{u}|\theta) = -\frac{n^2}{\sigma_{DP}^2}||\tilde{u} - \mu(\theta)||_2^2 + \mathrm{constant}, \tag{21}$$

so maximising the log-likelihood is equivalent to minimising the PGM objective.

If we made a normal approximation instead of the law of large numbers approximation in (15), we would get

$$\tilde{u} \sim \mathcal{N}(\mu(\theta), \Sigma(\theta)/n + \sigma_{DP}^2 I/n^2), \tag{22}$$

so maximising the likelihood is still possible. Unlike PGM, this maximum likelihood objective includes the covariance $\Sigma(\theta)$. We leave any comparisons between maximising this objective and PGM to future work.

# E  Hyperparameters

**NAPSU-MQ**  The hyperparameters of NAPSU-MQ are the choice of prior, choice of inference algorithm, and the parameters of that algorithm. For the toy data experiment, we used the Laplace approximation for inference, which approximates the posterior with a Gaussian centered at the maximum aposteriori estimate (MAP). We find the MAP for the Laplace approximation with the LBFGS optimisation algorithm, which we run until the loss improves by less than $10^{-5}$ in an iteration, up to a maximum of 500 iterations. Sometimes LBFGS failed to converge, which we detect by checking if the loss increased by over 1000 in one iteration, and fix by restarting optimisation from a different starting point. We also restarted if the maximum number of iterations was reached without convergence. For almost all runs, no restarts were needed, and at most 2 were needed.

For the Adult experiment, we used NUTS [14]. We ran 4 chains of 800 warmup samples and 2000 kept samples. We set the maximum tree depth of NUTS to 12. We normalised the posterior using the mean and covariance from the Laplace approximation. For the Laplace approximation, we used the same hyperparameters as with the toy data set, except we set the maximum number of iterations to 6000.

For the prior, we used a Gaussian distribution with mean 0 and standard deviation 10 for all components, without dependencies between components, for both experiments.

**PGM and Repeated PGM**  PGM finds the $\mathrm{MED}_\theta$ parameters $\theta$ that minimise the $L_2$-error between the expected query values and the noisy query values. The PGM implementation offers several algorithms for this optimisation problem, but we found that the default algorithm (mirror descent) and number of iterations works well for both experiments.

**RAP**  RAP minimises the error on the selected queries of a continuous relaxation of the discrete synthetic dataset. After the optimal relaxed synthetic dataset is found, a discrete synthetic dataset is constructed by sampling. This gives two hyperparameters that control the size of the synthetic data: the size of the continuous dataset, and the number of samples for each datapoint in the continuous relaxation. We set the size of the continuous dataset to 1000 for both experiments, as recommended by the paper [2]. For the Adult data experiment, we set the number of samples per datapoint to 46, so that the total size of the synthetic dataset is close to the size of the original dataset. For the toy data experiment, we set the number of samples per datapoint to 50. The RAP paper [2] finds that much smaller values are sufficient, but higher values should only increase accuracy.

In both cases, we weight the synthetic datapoints by $\frac{n}{n_{Syn}}$ before the downstream logistic regression to ensure that the logistic regression does not over- or underestimate variances because of a different sample size from the original data.

RAP also has two other hyperparameters that are relevant in our experiments: the number of iterations and the learning rate for the query error minimisation. After preliminary runs, we set the learning rate at 0.1 for both experiments, and set the number of iterations to 5000 for the toy data experiment, and 10000 for the Adult data experiment.

**PEP**  PEP has two hyperparameters: the number of iterations used to find a distribution with maximum entropy that has approximately correct query values, and the allowed bound on the difference of the query values. The PEP implementation hardcodes the allowed difference to 0. We set the number of iterations to 1000 after preliminary runs for both experiments.

**PrivLCM**  PrivLCM samples the posterior of a Bayesian latent class model, where the number of classes in limited to make inference tractable. The model has hyperparameters for the prior, and the number of latent classes. We leave the prior hyperparameters to their defaults, and set the number of latent classes to 10, which the PrivLCM authors used in a 5-dimensional binary data experiment [28]. The remaining hyperparameter of PrivLCM is the number of posterior samples that are obtained. To keep the runtime of PrivLCM manageable, we set the number of samples to 500 after ensuring that the lower number of samples did not degrade the accuracy of the estimated probabilities for the joint distribution compared to using the default of 5000 samples.

# F   Toy Data Experiment Details

To demonstrate the necessity of noise-awareness in synthetic data generation, we measure the coverage of confidence intervals computed from DP synthetic data on a generated toy dataset where the data generation process is known. We test the existing algorithms PGM [26], PEP [20], RAP [2], PrivLCM [28] and our pipeline NA+MI, where data generation is implemented with NAPSU-MQ. The authors of PrivLCM also propose using multiple imputation [28], so we use Rubin's rules [29] with the output of PrivLCM.

The original data consists of $n = 2000$ datapoints of 3 binary variables. The first two are sampled by independent coinflips. The third is sampled from logistic regression on the other two variables with coefficients $(1, 0)$.

For all algorithms except PrivLCM, we use the full set of 3-way marginal queries released with the Gaussian mechanism. PrivLCM doesn't implement these, and instead uses all full sets of 2-way marginals, and a different mechanism, which is $(\epsilon, 0)$-DP [28] instead of $(\epsilon, \delta)$-DP like the other algorithms. We use the Laplace approximation for NAPSU-MQ inference, as it is much faster than NUTS and works well for this simple setting.

For the privacy bounds, we use $\delta = n^{-2}$, and vary $\epsilon$. We generate $m = 100$ synthetic datasets of size $n_{Syn} = n$ for all algorithms except RAP, where the synthetic dataset size is a function of two hyperparameters. We describe the hyperparameters in detail in Supplemental Section E.

The downstream task is inferring the logistic regression coefficients from synthetic data. We repeated all steps 100 times to measure the probability of sampling a dataset giving a confidence interval that includes the true parameter values.

Figure 1 shows the coverages, and Figure 3a shows the widths for the resulting confidence intervals. All of the algorithms apart from ours and PrivLCM are overconfident, even with very loose privacy bounds. Examining the confidence intervals shows the reason: PGM is unbiased, but it produces too narrow confidence intervals, while NAPSU-MQ produces wider confidence intervals. On the other hand, for $\epsilon > 0.25$, PrivLCM produces much wider and too conservative confidence intervals.

# G   Adult Experiment Details and Extra Results

We include the columns Age, Workclass, Education, Marital Status, Race, Gender, Capital gain, Capital loss, Hours per week and Income of the Adult dataset, and discard the rest to remove redundant columns and keep computation times manageable. We discretise Age and Hours per week to 5 buckets, and discretise Capital gain and Capital loss to binary values indicating whether their value is positive. The Income column is binary from the start, and indicates whether a person has an income $> \$50\,000$.

In the downstream logistic regression, we use income as the dependent variable, and Age, Race and Gender as independent variables. Age is transformed back to a continuous value for the logistic regression by picking the middle value of each discretisation bucket. We did not use all variables for the downstream task, as a smaller set of variables allows including the relevant marginals for synthetic data generation. The regularisation for the logistic regression is $l_2$ with a very small multiplier of 0.00001. When the regularisation is used, variances are estimated with bootstrapping using 50 bootstrap samples.

All of the Adult experiment figures, except Figures S5 and S6 use the small regularisation term. Figure 4 shows the results with the trick of removing large variance estimates $(> 10^3)$, and Figure S6 shows how many estimates were removed.

For the input queries, we include the 2-way marginals with Hours per week and each of the independent variables Age, Race and Gender and income, as well as the 2-way marginal between Race and Gender. The rest of the queries were selected with the MST algorithm [24]. For MST, we used $\epsilon = 0.5$, but we do not include this in our figures, as we focus on the synthetic data generation, not query selection. The selected queries are shown in Figure S1. The selection is very stable: in 100 repeats of query selection, these queries were selected 99 times.

We chose the number of generated synthetic datasets for NAPSU-MQ and the number of repeats for repeated PGM by comparing the results of the Adult experiment for different choices. The results

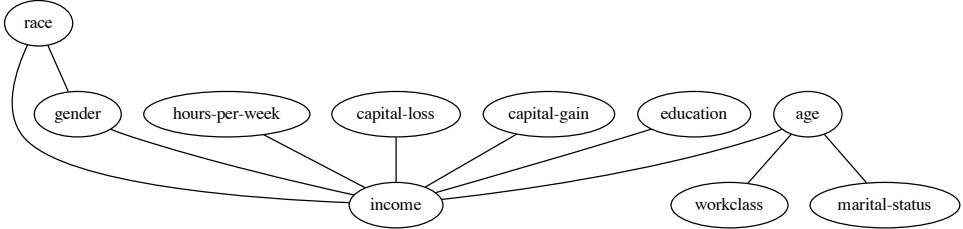

Figure S1: Markov network of selected queries for the Adult experiment. Each edge in the graph represents a selected 2-way marginal.

are shown in Figure S4 for NAPSU-MQ and Figure S3 for repeated PGM. We chose $m = 100$ for NAPSU-MQ because it produces the narrowest confidence intervals, and $m = 5$ repeats for repeated PGM because it had the best calibration overall.

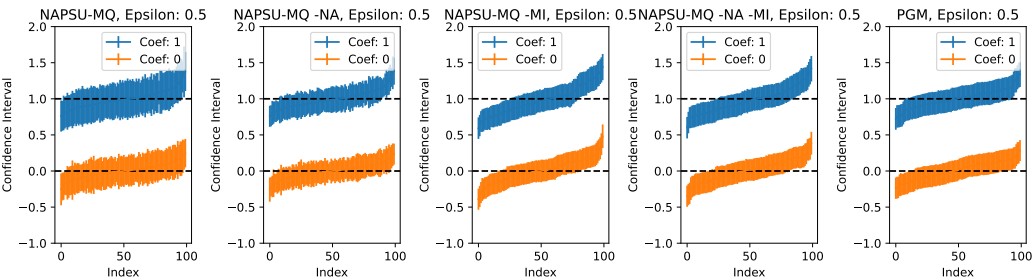

Figure S2: Ablation Study Confidence Intervals

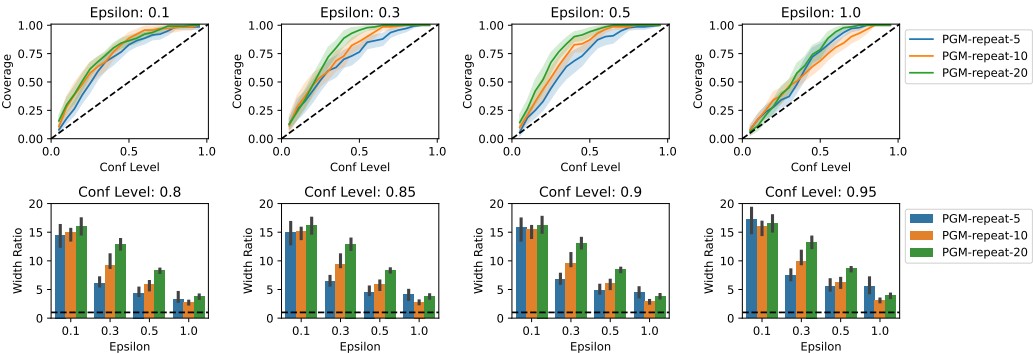

Figure S3: Comparison of different numbers of repetitions for repeated PGM on the Adult dataset with regularisation. We chose $m = 5$ repeats to represent repeated PGM in the main experiment, although the differences between the numbers of repeats are small.

## H  US Census Data Experiment

We conducted an additional experiment on US Census data from the UCI repository [27]. We limited the data to individuals who have served in the US Military, and picked 9 columns[2], most relating to military service. Even this subset of the data is large, with $n = 320\,754$. All columns are discrete, and have $10\,800$ possible values, much fewer than the Adult experiment.

---

[2] The columns are dYrsserv, iSex, iVietnam, iKorean, iMilitary, dPoverty, iMobillim, iEnglish and iMarital.

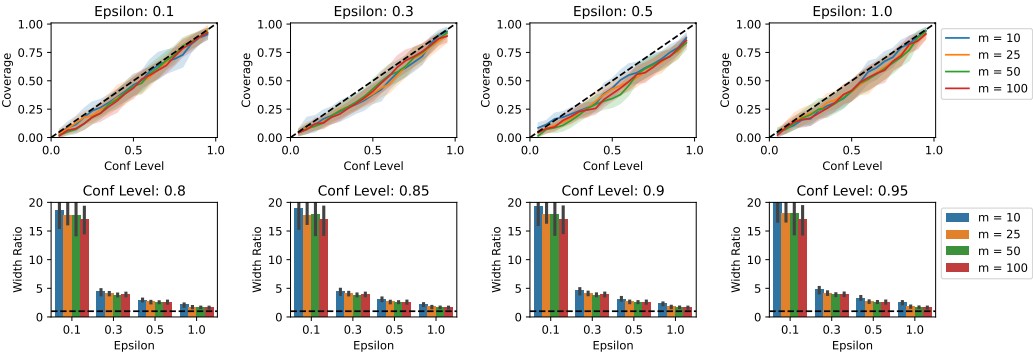

Figure S4: Comparison of different numbers of generated synthetic datasets for NAPSU-MQ on the Adult dataset with regularisation. The differences are small, but $m = 100$ synthetic datasets produces the narrowest intervals, so we chose it for the main experiment.

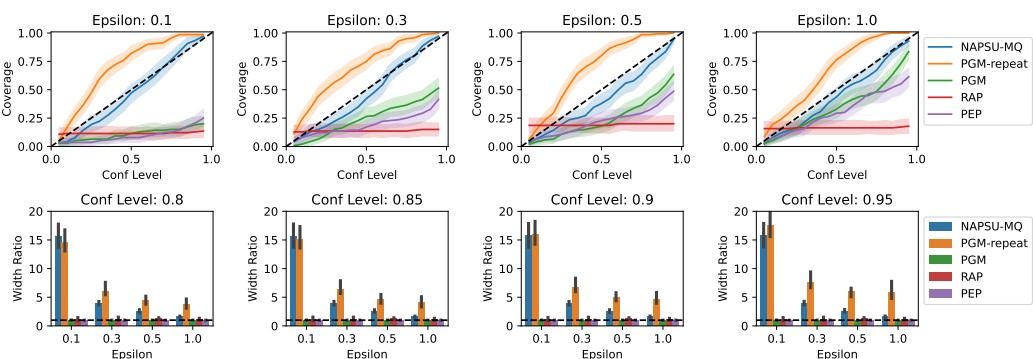

Figure S5: Results from the Adult data experiment with the trick of dropping large variances in the logistic regression instead of adding a small regularisation term. The results are almost identical to Figure 4, except for RAP, which suffers from the regularisation.

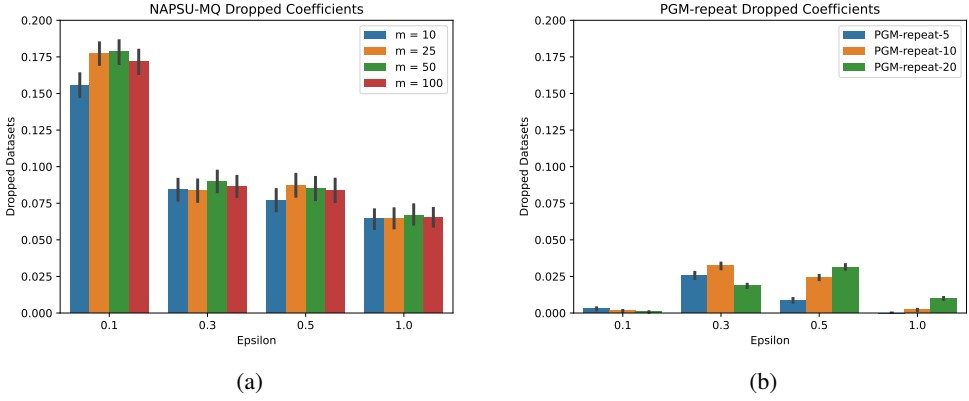

Figure S6: The fraction of coefficients dropped before Rubin's rules because of very high estimated variances from the downstream logistic regression in the Adult data experiment for NAPSU-MQ in (a) and PGM-repeat in (b).

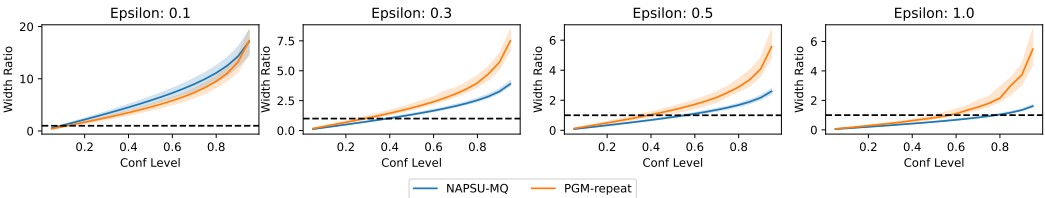

Figure S7: The tradeoff between the confidence level for DP confidence intervals and the width of the intervals on the Adult dataset with regularisation. The width ratio on the y-axis is with regards to the original 95% confidence interval, for all confidence levels, so the plot shows how much must the confidence level drop to obtain the same width from a DP confidence interval as a non-DP one. The horizontal line at $y = 1$ shows this point. For $\epsilon = 1$, the confidence level for NAPSU-MQ must be dropped to about 75%, and for PGM-repeat, it must be dropped to about 50%.

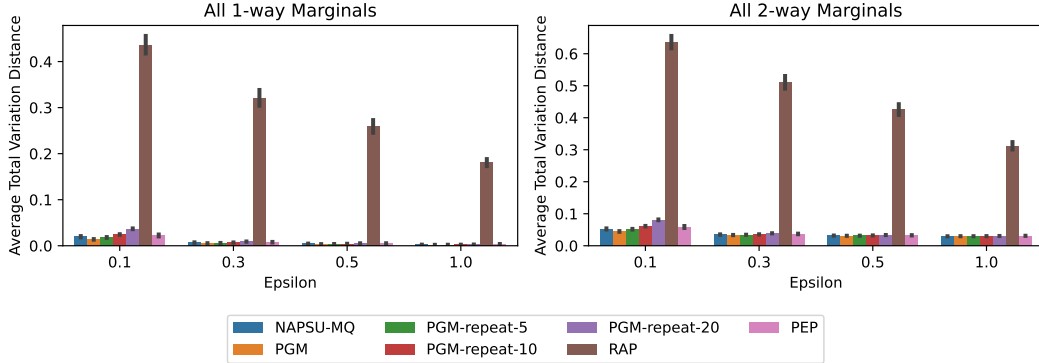

Figure S8: Comparison of marginal query accuracy for Adult data. NAPSU-MQ is almost as accurate as PGM for all values of $\epsilon$, and is on par with PGM-repeat. The panels show the average total variation distance of all 1-way marginal distributions (left) or all 2-way marginal distributions (right) between the original discretised data and synthetic data, averaged over 20 repeats. For NAPSU-MQ and PGM-repeat-$m$, the synthetic marginal distributions were estimated by averaging over $m$ synthetic datasets, with $m = 100$ for NAPSU-MQ.

As the downstream task, we use logistic regression with iPoverty as the dependent variable and iSex, iKorean, iVietnam and iMilitary as the independent variables. iPoverty has three categories, so we combine the two categories denoting people below the powerty line to make the dependent variable binary for the logistic regression, but not synthetic data generation.

As our queries we use 4 three-way marginals covering the independent and dependent variables, and 3 two-way marginals that include the other variables that are synthesised, but not included in the regression. As the published implementation of RAP [2] does not support a mix of two- and three-way marginals, we replace the two-way marginals with three-way marginals for RAP. As in the Adult experiment, we set $\delta = n^{-2}$, and vary $\epsilon$.

As in the adult experiment, we use $n_{Syn} = n$ for all algorithms except RAP. For PGM-repeat and NAPSU-MQ, we choose $m$ with a preliminary experiment. For NAPSU-MQ, we set $m = 100$, although the differences between the choices are not large. For PGM-repeat, we set $m = 10$. We set the other hyperparameters for all algorithms after testing runs to the same values used in the Adult experiment, except we increased the number of optimisation iterations for PGM to 5000 from the default of 1000, and dropped the number of NUTS chains to 2 and the number of warmup samples to 400 for NAPSU-MQ. We did not use the trick of dropping estimates with very high variances, or using very small regularisation in the logistic regression with the US Census data.

The results are shown in Figure S9. While PGM is calibrated with $\epsilon \geq 0.3$, it is severely overconfident with $\epsilon = 0.1$. This is likely caused by the large size of the dataset: at larger values of $\epsilon$, there is little noise compared to the large sample size, while at $\epsilon = 0.1$, the noise has a clear effect.

Table S1: Runtimes of each inference run for the Adult experiment. Does not include the time taken to generate synthetic data, or run any downstream analysis. The LA rows record the runtime for obtaining the Laplace approximation for NAPSU-MQ that is used in the NUTS inference, so the total runtime for a NAPSU-MQ run with NUTS is the sum of the LA and NUTS rows. Experiments were run on 4 CPU cores of a cluster.

| Algorithm | Epsilon | Mean | Standard Deviation |
|---|---|---|---|
| LA | 0.1 | 2 min 53 s | 18.5 s |
| | 0.3 | 3 min 53 s | 29.4 s |
| | 0.5 | 3 min 38 s | 35.0 s |
| | 1.0 | 3 min 25 s | 25.5 s |
| NUTS | 0.1 | 9 h 59 min 6 s | 6506 s |
| | 0.3 | 7 h 33 min 28 s | 2701 s |
| | 0.5 | 4 h 57 min 40 s | 3185 s |
| | 1.0 | 3 h 51 min 34 s | 1274 s |
| PEP | 0.1 | 6 min 50 s | 25.4 s |
| | 0.3 | 7 min 18 s | 31.2 s |
| | 0.5 | 7 min 0 s | 33.1 s |
| | 1.0 | 7 min 7 s | 33.7 s |
| PGM | 0.1 | 15 s | 0.5 s |
| | 0.3 | 17 s | 1.5 s |
| | 0.5 | 15 s | 0.4 s |
| | 1.0 | 15 s | 0.6 s |
| PGM-repeat-10 | 0.1 | 2 min 35 s | 3.3 s |
| | 0.3 | 2 min 53 s | 13.0 s |
| | 0.5 | 2 min 37 s | 5.0 s |
| | 1.0 | 2 min 36 s | 4.4 s |
| PGM-repeat-20 | 0.1 | 5 min 15 s | 10.9 s |
| | 0.3 | 5 min 58 s | 28.4 s |
| | 0.5 | 5 min 10 s | 10.2 s |
| | 1.0 | 5 min 13 s | 12.6 s |
| PGM-repeat-5 | 0.1 | 1 min 17 s | 2.7 s |
| | 0.3 | 1 min 28 s | 6.7 s |
| | 0.5 | 1 min 18 s | 2.6 s |
| | 1.0 | 1 min 18 s | 1.9 s |
| RAP | 0.1 | 32 s | 2.4 s |
| | 0.3 | 34 s | 2.2 s |
| | 0.5 | 32 s | 2.1 s |
| | 1.0 | 31 s | 2.1 s |

NAPSU-MQ and PGM-repeat are able to produce calibrated results at $\epsilon = 0.1$. Of these, NAPSU-MQ produces clearly narrower confidence intervals for both values of $\epsilon$.

Figure S10 shows the accuracies of the produced synthetic datasets on all 1-way and 2-way marginal queries for the algorithms. As with the Adult dataset, shown in Figure S8, NAPSU-MQ is almost as accurate as PGM, and is equally accurate as PGM-repeat. For $\epsilon \geq 0.5$, all of the aforementioned algorithms are almost as accurate. RAP and PEP are nowhere close to these algorithms in accuracy, having errors that are several times larger than the other algorithms.

For some reason, PEP fails completely with this dataset. We are not sure what causes this, as the algorithm should work in this setting as well as it did with the Adult dataset, and the size of the dataset should not be an issue.

The runtimes for each algorithm are shown in Table S2. The difference between PGM-repeat and NAPSU-MQ is much smaller than in the Adult data experiment, but is still large.

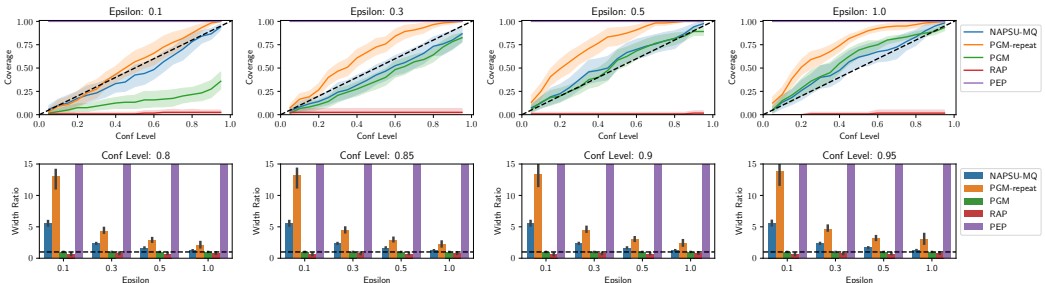

Figure S9: Results for the US Census experiment, showing that only NAPSU-MQ and PGM-repeat are calibrated for all values of $\epsilon$, and NAPSU-MQ produces significantly narrower confidence intervals than PGM-repeat. Like Figure 4, the top row shows the mean coverage over all coefficients and 20 runs for different confidence levels. The bottom row shows median confidence interval widths divided by real data confidence interval widths.

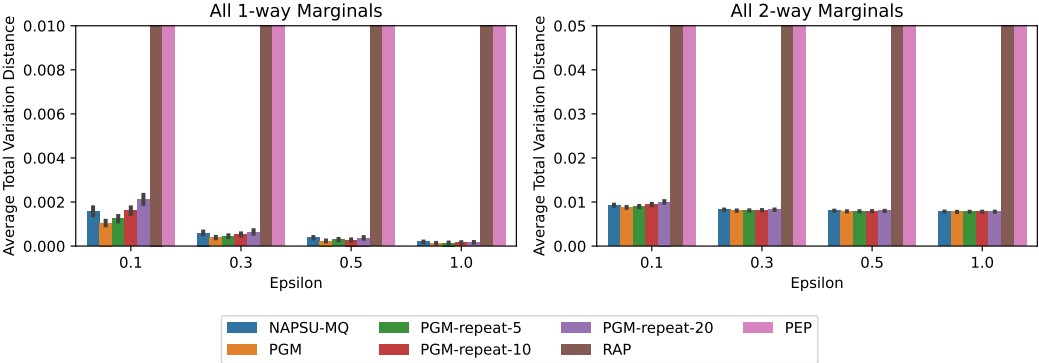

Figure S10: Comparison of marginal query accuracy for US Census data. NAPSU-MQ is almost as accurate as PGM for all values of $\epsilon$, and is on par with PGM-repeat, as with the Adult data in Figure S8. The panels show the average total variation distance of all 1-way marginal distributions (left) or all 2-way marginal distributions (right) between the original discretised data and synthetic data, averaged over 20 repeats. For NAPSU-MQ and PGM-repeat-$m$, the synthetic marginal distributions were estimated by averaging over $m$ synthetic datasets, with $m = 100$ for NAPSU-MQ. RAP and PEP have average total variation distances over 0.1 for both 1-way and 2-way marginals for all values of $\epsilon$.

Table S2: Runtimes of each inference run for the US Census experiment. Does not include the time taken to generate synthetic data, or run any downstream analysis. The LA rows record the runtime for obtaining the Laplace approximation for NAPSU-MQ that is used in the NUTS inference, so the total runtime for a NAPSU-MQ run with NUTS is the sum of the LA and NUTS rows. Experiments were run on 4 CPU cores of a cluster.

| Algorithm | Epsilon | Mean | Standard Deviation |
|---|---|---|---|
| LA | 0.1 | 2 min 2 s | 76.8 s |
| | 0.3 | 1 min 35 s | 23.8 s |
| | 0.5 | 1 min 51 s | 44.0 s |
| | 1.0 | 1 min 51 s | 44.8 s |
| NUTS | 0.1 | 3 h 32 min 25 s | 2836 s |
| | 0.3 | 1 h 57 min 45 s | 989 s |
| | 0.5 | 1 h 31 min 15 s | 951 s |
| | 1.0 | 1 h 8 min 6 s | 477 s |
| PEP | 0.1 | 17 s | 0.6 s |
| | 0.3 | 18 s | 1.0 s |
| | 0.5 | 17 s | 0.4 s |
| | 1.0 | 17 s | 0.5 s |
| PGM | 0.1 | 1 min 57 s | 2.8 s |
| | 0.3 | 1 min 58 s | 3.2 s |
| | 0.5 | 1 min 57 s | 4.2 s |
| | 1.0 | 1 min 57 s | 3.2 s |
| PGM-repeat-10 | 0.1 | 19 min 22 s | 28.0 s |
| | 0.3 | 19 min 18 s | 21.9 s |
| | 0.5 | 19 min 36 s | 33.9 s |
| | 1.0 | 19 min 25 s | 22.0 s |
| PGM-repeat-20 | 0.1 | 38 min 38 s | 50.2 s |
| | 0.3 | 38 min 59 s | 37.5 s |
| | 0.5 | 38 min 57 s | 40.5 s |
| | 1.0 | 38 min 39 s | 74.7 s |
| PGM-repeat-5 | 0.1 | 9 min 45 s | 17.6 s |
| | 0.3 | 9 min 38 s | 8.9 s |
| | 0.5 | 9 min 45 s | 12.8 s |
| | 1.0 | 9 min 43 s | 9.8 s |
| RAP | 0.1 | 28 s | 2.4 s |
| | 0.3 | 28 s | 2.0 s |
| | 0.5 | 27 s | 1.2 s |
| | 1.0 | 27 s | 3.5 s |

