# OpenReview forum: "Noise-Aware Statistical Inference with Differentially Private Synthetic Data"
_NeurIPS.cc/2022/Workshop/SyntheticData4ML — Neurips 2022 SyntheticData4ML_

### Official Review · Reviewer_Q63F · 2022-10-10
**Interesting problem, but applicability seems too limited**

**Rating:** 4
**Confidence:** 3

**Review:**

Studying uncertainty of DP synthetic data is an interesting problem, and the paper is well-written and presented. However, the proposed method’s applicability seems too limiting for practical use.

**Weaknesses, questions and suggestions**

1. Binary data. The method focuses on binary datasets, which greatly limits its applicability to most machine learning problems. In the experiments the Adult dataset is used, which contains both categorical and continuous and is hence discretized by the authors. It would be good to understand to what extent discretisation is used, to further understand what information is lost in this process and to what extent the method can be applied to real-world datasets.

2. Number of features. It is mentioned that the runtime is very large and that this warrants reducing the number of features in the Adult dataset to just 10 features. For understanding the method’s practical use, it would be good to show how it scales, either theoretically or experimentally.

3. Choosing marginal queries. Line 147, “we pick 2-way marginals that are relevant for the downstream task”. It is unclear how this would exactly work, and this itself is a limitation that is not explicitly mentioned (e.g. if the downstream task is not known by the data holder).

4. Scope and positioning. Synthetic data generation has become a broad field. The scope of the work can be improved throughout the paper, e.g. by more clearly labelling limitations and contributions in the introduction instead of waiting until Section 3. Additionally, from the perspective of a deep generative modelling researcher, the idea of using marginal queries for synthetic data is not well motivated, which could be aided by clearer positioning.

5. Use cases unclear. It would be good to have specific use cases for when this method is both applicable---in the sense that the data is binary, small number of features, downstream task is known---, but also relevant, e.g. easier for a data holder compared to simply running the downstream analysis themselves.

---

### Official Review · Reviewer_EDZZ · 2022-10-18

**Rating:** 7
**Confidence:** 3

**Review:**

The paper proposes a method for a noise-aware generation of synthetic datasets in order to account for the noise introduced by DP-mechanisms. Using this approach to generate multiple datasets allows users to combine the analysis results of each dataset using Rubin's rules, providing a posterior of the analysis results in the form of a t-distribution.

The paper is well-written, the idea is sound and the empirical results support the claims. Thus I vote for acceptance.

Detailed comments:
- In Alg. 1, \widetilde{S} is sampled with mean s, instead of according to Eq. 3. The text suggests that rather Eg. 3 is used to sample \widetild{s}. Could you clarify?
- regarding Eq. 3: how close are samples drawn according to Eq. 3 to samples drawn with mean s in practice?
- for the paper to be self-contained, it would be great to get an explanation of Rubin's rules.

---

### Official Review · Reviewer_hRCW · 2022-10-19
**Tighter confidence interval estimation for DP synthetic data utilizing methods like Rubin rules**

**Rating:** 7
**Confidence:** 4

**Review:**

The authors tackle the problem of providing uncertainty estimates for model parameters utilizing the synthetic data and showing them to reliable as using the real datasets.

Pros:
(A) They take advantage of generating multiple datasets (data holder) and use Rubin rules (data analyst) to account for the uncertainty added by synthetic data. They build on existing work from Bernstein and Sheldon to build noise aware synthetic data from marginal queries.
(B) Experimental results shows the tighter confidence intervals than based on the competitors including PrivLCM.

Cons:
(1) Not clear how to select m for the number of synthetic datasets and if that causes additional overhead due to extra space reasons?
(2) The setting is quite limited to discrete tabular data as mentioned in the paper.
Overall, a good paper that tackles the downstream questions arising from utilizing synthetic data as a proxy for real data.

---

### Meta-Review · Area_Chair_T2yj · 2022-10-18

**Recommendation:** Accept